# A Review of Models and Algorithms for Surface-Underground Mining Options and Transitions Optimization: Some Lessons Learnt and the Way Forward

Bright Oppong Afum  and Eugene Ben-Awuah *

Mining Optimization Laboratory (MOL), Bharti School of Engineering, Laurentian University, Sudbury, ON P3E 2C6, Canada; bafum@laurentian.ca

* Correspondence: ebenawuah@laurentian.ca

**Abstract:** It is important that the strategic mine plan makes optimum use of available resources and provides continuous quality ore to drive sustainable mining and profitability. This requires the development of a well-integrated strategy of mining options for surface and/or underground mining and their interactions. Understanding the current tools and methodologies used in the mining industry for surface and underground mining options and transitions planning are essential to dealing with complex and deep-seated deposits that are amenable to both open pit and underground mining. In this study, extensive literature review and a gap analysis matrix are used to identify the limitations and opportunities for further research in surface-underground mining options and transitions optimization for comprehensive resource development planning.

**Keywords:** strategic mining options optimization; mathematical programming models; transition depth; open pit-underground mining; resource development planning



## 1. Introduction

Surface mining is known to be relatively highly productive, very economic, and safer for workers compared to underground mining for most suitable deposits. However, recent evolution in environmental regulations and societal expectations may result in the development of small, high-grade deposits by shallow open pits (OP) or in the establishment of high-grade underground (UG) mines in place of extensive OP operations [1]. Optimizing the extraction of a mineral deposit in the presence of both surface mining methods and UG mining methods result in the most economic decision generated by identifying the best mining option for the deposit. In resource development planning, optimizing resource exploitation depends largely on the mining option used in the extraction. The term mining options optimization has been used by researchers and professionals to refer to the initiatives or choices undertaken in the extractive industry to expand, change, defer, abandon, or adopt strategies for a mining method(s) and sometimes investment opportunities; based on changing economics, technology, or market conditions [2–10]. For mineral deposits with orebodies that extend from near surface to several depths, such orebodies are amenable to different variations of either open pit mining, underground mining, or both.

Some studies have been conducted to solve the surface-underground mining options and transitions optimization (SUMOTO) problem. These studies have focused on determining the transition depth and the resulting production schedules for the OP and UG mining operations using simplified optimization frameworks. These models do not extensively address the multi-objective optimization nature of the SUMOTO problem, and do not formulate the problem with a complete description of the practical mining environment. Specifically, the existing models do not incorporate essential developmental infrastructure such as primary and secondary mine accesses, ventilation requirement, and geotechnical support and reinforcement in the optimization framework. Results from these models

often lead to localized optimal solutions or biased solutions that are usually impractical to implement in the mining environment [2,3,11–24].

Existing optimization algorithms used in attempting the mining options problem include the Lerchs-Grossman (LG) algorithm, Seymour algorithm, floating cone technique, network flows, dynamic programming, neural network, theory of graphs, and mathematical formulations [25]. Some authors have studied the surface-underground mining options and transitions optimization problem with available commercial software packages including Surpac Vision, Datamine's NPV Scheduler, Whittle Four-X, Geovia MineSched, integrated 3D CAD systems of Datamine, Vulcan, MineScape, MineSight, Isatis, XPAC, Mineable Reserve Optimizer (MRO), Blasor pit optimization tool, COMET cut-off grade and schedule optimizer, and Datamine Studio 3 [7,9,17,26,27]. The techniques used by these authors are not generic but mostly scenario based and often lead to localized optimization solutions.

Although Bakhtavar, Shahriar, and Oraee [3] employed a heuristic algorithm to compare economic block values computed for both open pit and underground mining on a depth flow basis to solve the SUMOTO problem, results from the heuristic algorithm do not offer a measure of optimality as it is the case in mathematical programming optimization. Notable authors that used mathematical programming to solve the mining transition problem limit their model to the determination of transition depth and block extraction sequence for the open pit and underground mining operations [2,4,8,9,11–15,17–19,24,25,28,29]. Similarly, other authors have developed stochastic mathematical programming models to solve the surface-underground mining options and transitions optimization problem. They focused on determination of the transition depth in 2D environment, and do not incorporate other essential underground mining constraints such as primary and secondary development, ventilation shaft development, and geotechnical requirements for the development openings and stopes in the optimization framework [7,20,30,31]. This is because the optimization of underground mines is computationally complex [32] and integrating it with open pit mining makes it more challenging [33].

The positioning of the required crown pillar thickness in the SUMOTO problem is key to the operations of such mines. Some authors pre-selected the depth of the crown pillar (transition depth) before evaluating portions above the crown pillar for open pit mining and portions below the crown pillar for underground mining [4,18,25,27,28]. This may lead to suboptimal solutions and will require evaluating multiple crown pillar locations in a scenario-based approach. A few authors have attempted to incorporate the positioning of the crown pillar in the optimization process [14,24,30,31,34,35]. Their models were good improvements over previous works but were missing some constraints such as the ventilation requirement and rock strength properties required for practical implementation. The transition from OP to UG mining is a complicated geomechanical process which requires the consideration of rock mass properties [36,37].

Bakhtavar [12] reviewed the combined open pit with underground mining methods for the past decade and noticed that the transition problem has been implemented in either simultaneous or non-simultaneous modes. He asserts that non-simultaneous mode of combined mining is more acceptable because large-scale underground caving methods with high productivity and low costs can be used. However, in simultaneous mode, horizontal and vertical slices underhand cut and fill with cemented backfill is more feasible to be used with OP mining. Afum, Ben-Awuah, and Askari-Nasab [35] implemented a mathematical programming model that allows the optimization approach to decide whether the mineral deposit should be exploited with either simultaneous, non-simultaneous, sequential, or any of these combinations thereof.

Most existing models in general do not include the requirements of essential underground mining infrastructure such main access to the underground mine (shaft or decline or adit development), ventilation development, operational development (levels, ore and waste drives, crosscuts), and necessary vertical development (ore passes, raises). Equally, these existing models do not incorporate rock strength properties in the SUMOTO problem.

Although these essential infrastructure and geotechnical characteristics of the rock formation are significant to underground mining operations, their added complexities make it difficult to be included in the SUMOTO models. According to Bullock [38], mine planning is an iterative process that requires looking at many options and determining which, in the long run, provides the optimum results. Using such iterative process could lead to some inferior or sub-optimal solution(s) that do not constitute the global optimal solution.

In summary, this paper reviews relevant literature on algorithms and models for the SUMOTO, identifies gaps and opportunities that can be explored for further research and implementation in the mining industry, and further introduces the significance of employing mathematical programming for planning resources amenable to both options. Figure 1 is a schematic representation of the surface–underground mining options and transitions planning problem for deposits amenable to both mining options.

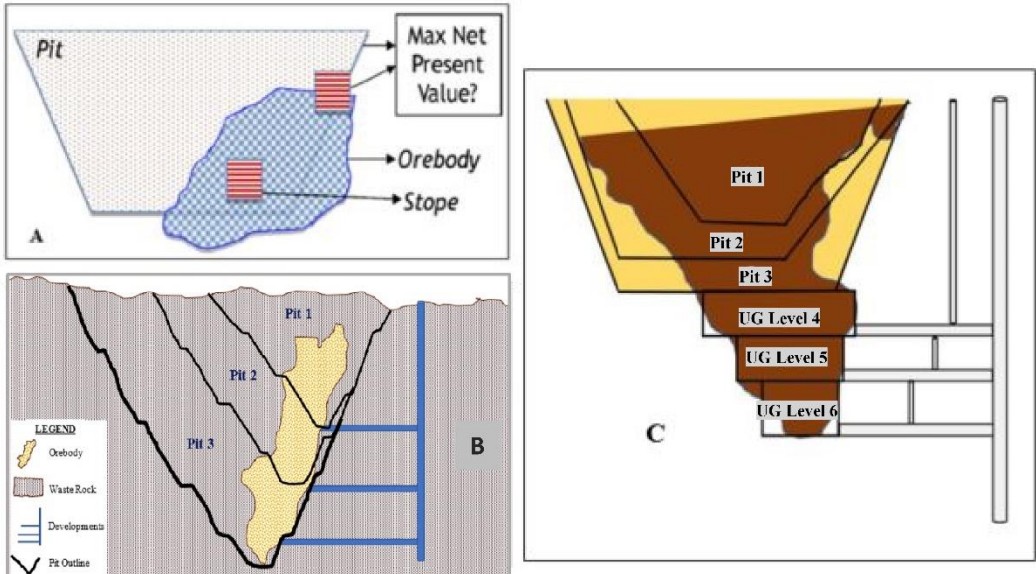

**Figure 1.** Schematic representation of the surface–underground mining options and transitions planning problem. (**A**) illustrates evaluation of an orebody to generate maximum net present value (NPV) depending on how each mining block is extracted; either through open pit mining, open stope extraction, or both. (**B,C**) demonstrate the extraction of a mineral resource by open pit (OP), underground (UG), or both open pit and underground (OPUG) mining for optimum resource development planning. ((**A**)—Ben-Awuah, Otto, Tarrant, and Yashar [4]; (**B**)—Afum, Ben-Awuah, and Askari-Nasab [34]).

## 1.1. Classification of Mining Methods (Mining Options)

Mining is defined as the process of exploiting a valuable mineral resource naturally occurring in the earth crust [39,40]. The extraction of mineral resources from the earth crust is classified broadly into two; surface mining and UG mining. In surface mining, all the extraction operations are exposed to the atmosphere while in UG mining, all the operations are conducted in the bosom of the earth crust. The main objective of a mineral project development is the maximization of investment returns; the "golden rule" of mining or the investor's "law of conservation" [41]. Therefore, adopting the best mining option that maximizes the project's value is a requirement to the establishment of a successful mine. Planning a surface mine is often simpler compared to an underground mine because there are broad similarities between different variations of surface mining as opposed to the variations of underground mining. Thus, planning an underground mine is necessarily complicated by the availability of many different types and variations of mining systems [37]. These surface and underground mining variations are also generally

referred to as classes of mining methods. The classification of surface mining methods and underground mining methods are respectively illustrated in Figures 2 and 3.

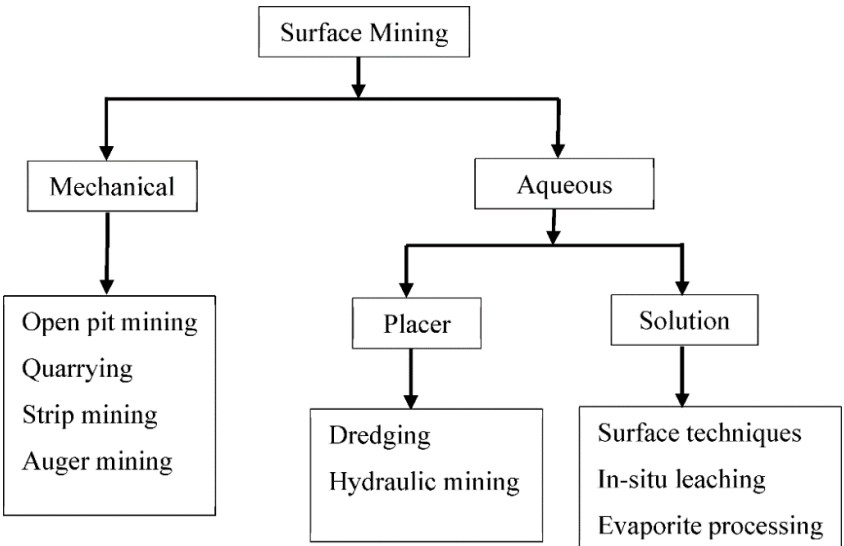

**Figure 2.** Classification of surface mining methods.

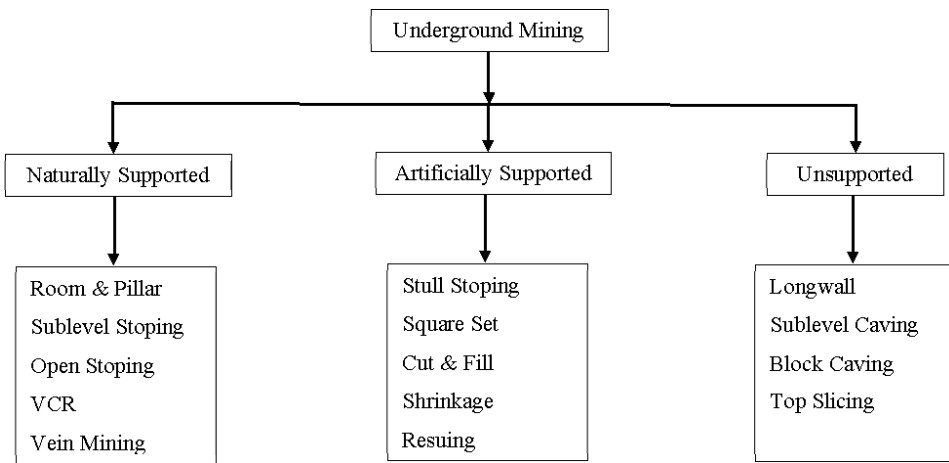

**Figure 3.** Classification of underground mining methods.

Surface mining methods are broadly classified into mechanical and aqueous extraction methods (Figure 2). Mechanical surface mining methods include open pit mining, quarrying, strip mining, and auger mining, while aqueous surface mining methods include placer mining and solution mining. Placer mining includes dredging and hydraulic mining while solution mining includes surface techniques such as in-situ leaching and evaporite processing. Based on the rock formation strength, UG mining methods are broadly classified into naturally supported methods, artificially supported methods, and unsupported methods (Figure 3). Naturally supported mining methods include room and pillar, sublevel stoping, open stoping, vertical crater retreat (VCR), and vein mining. Artificially supported mining methods include stull stoping, square set, cut and fill, shrinkage, and resuing while unsupported methods include longwall, sublevel caving, block caving, and top slicing. According to Nelson [1], some of the factors that must be considered when choosing between surface or underground mining methods include:

1.  Extent, shape, and depth of the deposit;
2.  Geological formation and geomechanical conditions;
3.  Productivities and equipment capacities;

4. Availability of skilled labor;
5. Capital and operating costs requirements;
6. Ore processing recoveries and revenues;
7. Safety and injuries;
8. Environmental impacts, during and after mining;
9. Reclamation and restoration requirements and costs;
10. Societal and cultural requirements.

*1.2. Mineral Projects Evaluation*

When a mineral deposit is discovered, several evaluations are conducted towards the project's viability. The evaluation methods used are broadly grouped into two: positive evaluation methods and normative evaluation methods [42]. Positive evaluation methods assess the quantity and quality of the mineral project while normative evaluation methods assess the social and ethical values of the mineral project. Positive evaluation methods deal with investigations related to the geology of the formation, technology required to develop the deposit, investment decisions such as net present value (NPV), internal rate of return (IRR), and options valuations, and financial evaluation to show how funds will be raised and repaid in future. This research focuses on positive evaluation methods for mineral projects with the exception of how funds are raised and repaid for the project.

A mineral property could be described as being at early-stage or advanced-stage exploration, development, defunct, dormant, or production stage [43]. Three types of studies are undertaken according to the stage of life of the mineral project under evaluation. These studies are scoping study, prefeasibility study, and feasibility study [44,45]. Scoping study is a preliminary assessment of the technical and economic viability of a mineral property while a prefeasibility study is a comprehensive study of the viability of the mineral project subject to operational constraints at which the preferred UG mining method or OP mining arrangement is established, including an effective mineral processing method. Feasibility study, on the other hand, is a comprehensive study on the design and cost of the selected mining option for developing the mineral project. Usually, the confidence level associated with a prefeasibility study is lower than a feasibility study while the confidence level for a scoping study is also lower than a prefeasibility study.

The outcome of a prefeasibility study on a mineral property is a mineral reserve that is profitable. Thus, the mineral resource is technically and economically evaluated and if it is profitable, it becomes a mineral reserve otherwise it remains a mineral resource until prevailing factors (commonly referred to as modifying factors) become favorable [44]. The details of the evaluation studies for a mineral project depends mainly on the stage of life of the mine and the prevailing regulatory requirements of the region. These detailed evaluation considerations may include geological and geostatistical modeling, geotechnical investigation, mining optimization studies, cost-benefit analysis, equipment selection, rock transportation studies, rock stability and slope requirement assessment, crown pillar location investigation, blasting and fragmentation studies, environmental baseline studies, environmental management and impact studies, and mine closure and reclamation studies. During prefeasibility studies, typical technical and economic considerations include geological, geostatistical, and mining optimization investigations. These investigations primarily define the spatial grade distribution of the deposit, and uncovers the size, shape, depth (extent), and orientation of the deposit, and further validate the profitability associated with the mining strategy.

Identifying the preferred mining option (whether OP or UG or both) during prefeasibility studies on a mineral property involves strategic optimization analysis. This analysis ensures much value is attained during the implementation of the resource development plan. When the mineral resource is closer to the earth surface (sometimes with significant outcrop), OP mining evaluation studies are outrightly conducted. When the mineral resource is deeply buried in the earth crust (with no significant outcrop or presence near the earth surface), UG mining evaluation studies are conducted. However, when the

mineral deposit is significantly closer to the earth surface and, also, deeply buried in the earth crust, the deposit becomes amenable to both OP and UG mining. In such cases, the portion of the orebody near the earth surface is evaluated to be exploited by OP mining method to produce early revenue, while the deeper portion is either outrightly evaluated for UG mining or the evaluation is deferred for later years in the future [7]. In general, OP mining methods are characterized by relatively low mining and operating costs, high stripping ratio, and extended time in accessing the mineral ore [4,46] while UG mining is characterized by high mining and operating costs, high-grade ore, and earlier times in retrieving the ore [31,47–50].

In most cases, for mineral deposits amenable to OPUG mining, the UG mining evaluation is not undertaken during prefeasibility studies but conducted in later years when the OP mine stripping ratio increases towards the critical limit. The effect of this traditional evaluation approach is an increased overall mining cost and a potential loss of financial benefits [51,52]. Similarly, where UG mining commences at the onset of the mineral project, and later converts or transitions to OP mining, significant financial loss can occur if the global mining strategy was not defined for the entire mineral deposit during prefeasibility studies. Historically, some mining companies that transitioned from OP to UG or vice versa include Telfer, Golden Grove, and Sunrise Dam in Australia [53]; Grasberg mine in Indonesia [54]; Akwaaba and Paboase mines of Kinross Chirano Gold Mine in Ghana [55]. In addition, Lac Des Iles mine located in Canada and Newmont Ahafo mine in Ghana both operate OPUG mining operations.

The extraction evaluation of such mineral resources amenable to OPUG mining is referred to in this research as mining options optimization [2,6,10,15]. The importance of rigorously assessing the economics for a mineral deposit extraction, before deciding on the mining option to adopt is therefore essential to mineral resource planning. This may ensure important decisions of canceling a major OP pushback and transitioning to UG mining or vice versa is known at the onset of the mining project [3,52].

## 2. Evaluation Techniques for Mining Options and Transitions Planning

The outcome of an evaluation study for a mineral deposit amenable to OPUG mining includes the optimal mining option, strategic extraction plan, and a transition depth or location. The variations of the mining option are independent OP mining, independent UG mining, concurrent OP and UG mining, OP mining followed by UG mining, and UG mining followed by OP mining. The strategic extraction plan includes the sequences of rock extraction and the determination of life of mine and transition depth. The transition depth defines the location or position of the crown pillar. The extraction strategy when OPUG mining is preferred could either be sequential mining or parallel mining or both [47]. Respectively, other researchers used the terms simultaneous or non-simultaneous or combined OPUG mining to refer to these same mining options [30].

Sequential mining is when the mineral deposit is continuously extracted by an independent OP mining method(s) until the pit limit is completely mined out before being followed by UG mining method(s), while parallel mining is when the mineral deposit is simultaneously or concurrently extracted by OP and UG mining in the same period or time. Transitioning is the main challenge for OPUG mining projects due to the complexity and implications of where and when to position the crown pillar (or identify the transition depth) [7] in the presence of various mining constraints. Over the years, five fundamental approaches have been used to determine the transition point or location of the crown pillar [24,47,56,57]. These techniques are (1) biggest economic pit, (2) incremental undiscounted cash flow, (3) automated scenario, (4) stripping ratio, and (5) opportunity cost analyses.

For the biggest economic pit approach, the mineral resource is primarily evaluated for OP mining. When the OP mining limit is obtained, the portion of the mineral resource falling outside the OP outline is evaluated for UG mining. The biggest economic pit is the simplest and most commonly used traditional approach for evaluating a mineral resource

amenable to OPUG mining options. For the biggest economic pit, the pit usually terminates when the marginal cost of waste stripping outweighs the marginal revenue obtained from processing additional amounts of ore.

In the case of the incremental undiscounted cash flow approach, the marginal OP profit from the mineral project per depth is evaluated and compared to the marginal UG profit. Due to increasing cost of stripping waste per depth, there is a point where the marginal OP profit is lower than the marginal UG profit. This depth is the transition point which then acts as the crown pillar during transition. This transition depth is typically shallower compared to the largest economic pit [47]. This method assumes that UG mining profits do not depend on the depth of operation, therefore, there will be a point where the marginal profits from UG mining operation will exceed that from OP mining operation.

The automated scenario analysis approach accounts for discounting unlike the incremental undiscounted cash flow approach. It is based on the premise that, per an equivalent unit of throughput, UG mines are characterized by high cut-off grades and therefore have higher cash flow compared to OP mines for the same throughput. The approach is implemented by compiling schedules for OP and UG mining and comparing the computed NPV for each potential transition point. Thus, a set of transition points are evaluated and the OPUG mining arrangements that offers the highest NPV is selected for further analysis and design [47]. This method is time consuming and complex.

The stripping ratio analysis features the use of allowable stripping ratio (ASR) planned by mine management for the OP mine and overall stripping ratio (OSR) computed per depth for the OP mine to determine the transition depth [56,57]. The stripping ratio is expressed by this relation with emphasis on exploitation cost of 1 tonne of ore in UG mining and in OP mining, as well as, removal cost of waste in relation to 1 tonne of ore extracted by OP mining. As OP mining deepens, the stripping ratio usually increases, increasing the overall mining cost. An OSR is calculated and used to determine the breakeven point of the OP mine relative to its depth. The OP mine transitions to UG mine when OSR is equal to the ASR established by management of the mining project.

The opportunity cost technique is an extension of the LG algorithm that optimizes the OP ultimal pit while considering the value of the next best alternative UG mining option. This approach also employs the strength of the undiscounted cashflow technique and assumes that rock materal in the transition zone will be mined by UG method if not mined by OP method [24]. The methodology ensures that a minimum opportunity cost is achieved for the selected optimal mining option at the expense of the unselected mining option.

An evaluation technique that seeks to leverage the advantages of all these five fundamental approaches were recently introduced by Afum and Ben-Awuah [33]. This approach is referred to as the competitive economic evaluation (CEE) technique. The CEE process evaluates each block of the mineral deposit and economically decides: (a) blocks suitable for OP mining, (b) blocks suitable for UG mining, (c) unmined blocks, and (d) unmined crown pillar simultaneously. The CEE optimization strategy is an unbiased approach that provides fair opportunity to each mining block for selection by a mining option.

### 2.1. Notable Research on Mining Options and Transitions Planning Optimization

Historically, a cash flow and NPV based algorithm was introduced in 1982 to examine the open pit-underground (OP-UG) transition interface [21]. Subsequently, in 1992, Nilsson reviewed the previous 1982 algorithm and produced a new algorithm that considers the transition depth as a critical input for evaluating deposits amenable to OP and UG extraction [22]. An algorithm for determining the depth of transition was thereafter introduced by Camus [58]. This algorithm was presented based on the economic block values for OP and UG extraction methods. The technique involves the implementation of the OP extraction algorithm taking into consideration an alternate cost due to UG exploitation. In 1997, a model was developed by Shinobe [10] that enables the mine operator to determine the optimum time of conversion based on discounted cash flow (DCF) techniques, and cost estimation equations according to O'Hara and Suboleski [59].

The model assumed that the underground resources were confined, and their extraction is technically feasible. Whittle Programming Pty developed an applied approach referred to as quantified operational scenarios for interfacing OP-UG mining methods in the OP to UG transition problem [60].

In 2001 and 2003, an evaluation technique based on allowable stripping ratio with a mathematical form for the objective function was introduced by Chen, Li, Luo, and Guo [57] and Chen, Guo, and Li [56]. Volumes of ore and waste within the final pit limit were assumed to be a function of depth for determining the transition point. A heuristic algorithm based on economic block values of a two-dimensional block model that compares the total value when using OP mining for extracting a particular level to UG mining of the same level was developed for the mining options problem [15]. The algorithm was based on the fact that typical ore deposits showing significant outcrops can be potentially exploited by OP mining followed by UG mining. Thus, mining is completed at the initial levels by OP extraction methods while transitioning to UG mining as the operation deepens from the middle levels.

Until 2009, only a few available algorithms could solve the optimal transition depth problem with some limitations. Bakhtavar, Shahriar, and Oraee [2] developed a model for solving the transition depth problem by modifying Nilsson's algorithm. The model generated two different mining schedules for OPUG mining. Each mining method is employed to extract mining blocks on the same level in series. The incremental NPVs of the extraction for each level blocks are compared and if the incremental NPV of the OP mine is larger than that of the UG mine, the algorithm transcends by adding the next series of level blocks to the previously optimized mining schedule; and the incremental NPV of the OPUG mine is compared again. Evaluation results from the first level to the last level is monitored to identify the optimal transition depth (level) for the establishment of a crown pillar. The remaining portions of ore below the crown pillar were evaluated and extracted utilizing UG stoping method(s).

Another heuristic model based on the economic block values of OPUG extraction was developed by Bakhtavar, Shahriar, and Mirhassani [14]. This new model was an improvement over the previous model by factoring in the NPV achieved through the mining process. Thus, for any level of the block model, the computed NPV from OP mining operation is compared to the NPV derived from UG mining operation for the same level. Although the model solves the transition problem using some technical and economic parameters, it does not consider mining and processing capacities (equipment requirements), and uncertainties in the geological and geotechnical characteristics of the orebody. Uncertainties of ore grades were considered in subsequent mathematical programming frameworks for the transition problem [20,30]. However, the implementation of these existing models has always assumed the mining options and transitions planning scheme to be a stepwise process and hence implements their solution strategy as OP mining followed by UG mining either in parallel (simultaneous) modes or sequential (non-simultaneous) modes [4,18,20,24,30]. The challenge to this assumption also forms the basis of this research. Table 1 shows a matrix comparison of notable research on the OP-UG mining options optimization problem in the last decade.

**Table 1.** Notable research on the open pits (OP)-underground (UG) mining options optimization problem for the past decade.

| Name of Author(s) | | Whittle et al. | MacNeil & Dimitrakopoulos | King et al. | Ben-Awuah et al. | De Carli & de Lemos | Ordin & Vasil'ev | Dagdelen & Traore | Roberts et al. | Opoku & Musingwini | Bakhtavar et al. |
|---|---|---|---|---|---|---|---|---|---|---|---|
| Year | | 2018 | 2017 | 2016 | 2016 | 2015 | 2014 | 2014 | 2013 | 2013 | 2012 |
| Research Focus | | Transition depth & production schedule | Transition Depth-Instances | Transition Depth-Instances | Assessment of Transition Problem | Transition Depth-Instances | Transition Depth-Dynamic | Transition Depth | Assessment of Transition Problem | Transition Depth-Dynamic | Transition Depth |
| Approach Used | Model/Algorithm | Modification of maximum graph closure method | Stochastic Integer Model | Mixed Integer Linear Programming (MILP) | Mixed Integer Linear Programming (MILP) | | Dynamic Programming-lag, trend, nonlinear | MILP-OptiMine® used to optimize the transition problem | MILP-OP & UG | | (0–1) Integer Programming |
| | Software Application | | | | Evaluator | Studio 3 & NPV Scheduler | | OptiMine, Whittle, Studio 5D and EPS | Blasor for OP optimization; COMET for OP schedule; Evaluator for UG optimization | Whittle for OP optimization; XPAC for OP schedule; Datamine's MRO for UG optimization | |

**Table 1.** *Cont.*

| Name of Author(s) | | Whittle et al. | MacNeil & Dimitrakopoulos | King et al. | Ben-Awuah et al. | De Carli & de Lemos | Ordin & Vasil'ev | Dagdelen & Traore | Roberts et al. | Opoku & Musingwini | Bakhtavar et al. |
|---|---|---|---|---|---|---|---|---|---|---|---|
| Outputs/Performance/ Transition Indicators | NPV | Yes | Yes | Yes | Yes | Yes | Yes | Yes | Yes | Yes | Yes |
| | IRR | | | | Yes | Yes | | | | | |
| | Mining Recovery | | | | | Yes | | | | | |
| | Avg. ROM Grade | | | | | | | | Yes | Yes | |
| | Metal Price to Cost Ratio | | | | | | | | | Yes | |
| | Production Rate | | | | | | Yes | | Yes | Yes | |
| | Stripping Ratio | | | | | Yes | Yes | | | Yes | |
| | Mine Life | | | | | Yes | Yes | Yes | Yes | | |
| | Revenue | | | | | | Yes | | | | |
| | Mining Cost | | | | | | | | | Yes | |
| | Production Schedule | | OP & UG | | OP, UG & OPUG | | Standalone for OP & UG | Standalone for OP & UG | | | |
| Commodity | | | Gold | Confidential | Gold-silver-copper | Gold | Coal mining–kimberlite pipe | Gold | Iron | Gold-4 different deposits | Hypothetical |
| Notable Remarks | | Mining sequence not considered | NPV compared to deterministic model | Optimality gap not improved | | | | | | Deterministic not good | |

As seen in Table 1, previous research on the mining options and transitions problems mainly focuses on the determination of transition depth before optimizing the production schedule for the mining arrangement. Formulating a model that allows the optimizer to select the most suitable extraction strategy, including independent OP, independent UG, simultaneous OPUG, sequential OPUG, or combinations of simultaneous and sequential OPUG, to exploit any deposit under consideration will be a major addition to the mining industry. It is important to note that in general previous research employed NPV and feasible production schedule as major performance indicators for the developed models and algorithms.

## 2.2. Factors Influencing Mining Options and Transitions Planning

The factors influencing mining options and transitions planning are also referred to as transition indicators. These transition indicators mostly depend on the shape and size of the mineral deposit and quantity of high-grade ore present in the selective mining units. These indicators usually vary for different orebodies and commodities, and the philosophy of mine management. The indicators can be broadly grouped into geologic, operational, geotechnical, economic, and mine management strategy on global business economic outlook [7,61]. These transition indicators are key to the identification and development of the set of constraints required for optimizing the OP-UG mining options and transitions planning problem [4,9,25].

Some of the indicators to consider in OP-UG transition optimization are mining recovery, commodity price, mineral grade, cost of extracting ore and stripping waste, mining and processing capacities, and UG dilution [62]. Subsequent research highlighted some additional important parameters that must be considered during transition planning [19]. These include workforce requirement, shape and size of the orebody, and geotechnical properties of the rock formation. Economic parameters such as discounting rate of expected cash flows and OPUG mining costs, and primary factors including the competence of mine management, characteristics of the geology of the orebody, stripping ratio, productivity rate, and capital cost requirements for the UG mining option will affect the decision for OP-UG transition as well [9,63].

The OP-UG mining options and transitions planning problem becomes complex when critical indicators such as crown pillar positioning and other essential underground mining constraints including primary and secondary access development, ventilation development, and geotechnical requirements for the development openings and stopes are integrated into the optimization framework. These essential UG mining developments were previously not considered in the optimization process because of computational complexities and the difficulty of integrating them with open pit mining operations [32]. The importance of incorporating crown pillar positioning, geotechnical support activity sequencing, underground infrastructure development, and mine management strategy in OP-UG mining options optimization studies are essential to attaining realistic mine plans [9,30].

When investigating OP-UG mining options, it is important to model all constraints and factors that have direct and remote impact on the economic and technical feasibility of the project. The strategic production plan is generally subject to several constraints that enforce the extraction sequence, blending requirements, and mining and processing capacity requirements for practical implementation.

## 2.3. Crown Pillar and Rock Strength Considerations in Mining Options and Transitions Planning

A crown pillar is the horizontal part of a rock formation between the first upper stope of an underground mine and an open pit excavation. A crown pillar is often provided to prevent the inflow of water from the OP floor to the UG workings, while reducing surface subsidence and caving. Finding the most suitable location of the crown pillar in a combined mining method of OPUG operations is one of the most interesting problems for mining engineers today, especially when crown pillar must collapse [14]. It is however worthy to

note that crown pillar usage in the OPUG mine is not universal as sometimes it is desirable for the crown pillar to collapse during the life of mine [24].

The positioning of the required crown pillar in the surface-underground mining options and transitions planning problem is key to the operations of such mines. Some researchers pre-selected the depth of the crown pillar (transition depth) before evaluating portions above the crown pillar for OP mining and portions below the crown pillar for underground mining [4,18,25,27,28]. This may lead to suboptimal solutions and will require evaluating multiple crown pillar locations in a scenario-based approach. A few authors have attempted to incorporate the positioning of the crown pillar in the optimization process [24,33,34]. Their models were good improvements over previous works but were missing some constraints such as the ventilation requirement and rock strength properties required for practical implementation. The transition from open pit to underground mining is a complicated geomechanical process which requires the consideration of rock mass properties [35,36].

In OP to UG transition, the challenge of deformation, displacement, and rock formation stability need to be meticulously investigated. These will ensure their effect on production, worker and equipment safety, and the working environment of the UG operation are highly secured [64]. When the crown pillar thickness is large, considerable quantity of mineral deposit is lost but when the pillar is undersized and thin, the probability of pillar failure and instability of the mine is eminent [65]. According to Ma, Zhao, Zhang, Guo, Wei, Wu, and Zhang [35], ground movement and deformation study of OP mines such as geometrical, geomechanical, and analytical analyses are important in the transition study.

Optimizing the crown pillar dimensions and positioning is particularly significant in UG mining operations. Estimation of the optimum crown pillar thickness is a complicated study which involves the experience of the engineer and the use of numerical and empirical techniques [65]. Several parameters influence crown pillar stability. These parameters are broadly grouped into mining and geological [5]. The mining parameters include the crown pillar geometry; stope surroundings; supporting methods employed (including backfilling); sequence of mining operations and stress redistribution resulting from material extraction. The geological parameters include the strength and deformation characteristics, and inclination of the hanging wall, footwall, and orebody in general; geometry of the mineral deposits; virgin stress conditions and properties of the contact regions between the ore and country rock.

Crown pillar placement invariably defines the interface of the OP to UG transition. Appropriately defining a suitable location of the crown pillar is fundamental to the mining options optimization problem. Leaving an appropriate crown pillar thickness will minimize the destructive interference between the OP and UG mining operations, while maximizing ore recovery. The assumption of a uniform crown pillar of known height below the optimized pit bottom is common [14]. Subsequently, an ad-hoc branch-and-bound technique to exhaustively search the appropriate locations for crown and sill pillar placements before computing the relaxed linear programming (LP) model for the OPUG mining problem was developed by King, Goycoolea, and Newman [18]. A rounding heuristic was used to transform the relaxed LP solution into integer programming (IP) solution using satisfactory values of the objective function. The IP solution was used to terminate the several potential crown and sill pillar placements and hence decrease the computational time required. The researchers further concluded that, only 40 out of the over 3500 crown and sill pillar placement options had a relaxed LP objective function value greater than the best-known IP objective function value.

In a recent research, MacNeil and Dimitrakopoulos [20] pre-determined four possible crown pillar positions while evaluating a gold deposit for OPUG mining. Their approach resulted in four distinct transition points acting as potential candidates for a crown pillar. The size of the crown pillar however remained the same for each candidate, as the location is varied. This assumption does not support the geotechnical variability of the rock formation. According to King, Goycoolea, and Newman [18], the crown pillar is usually positioned by

industrial experts based on: (1) the optimal OP mining limit, or (2) the largest undiscounted profit resulting from the extraction technique for each 3-D discretization of the orebody and country rock. This is conducted after a detailed geotechnical assessment of the rock formation in the immediate vicinity of the crown pillar.

The stability and placement of the crown pillar or transition interface significantly affects the NPV of a mining project. Numerical simulation and machine learning are among the most effective techniques for studying the crown pillar stability [66–68]. The Fast Lagrangian Analysis of Continua (FLAC) software based on numerical modeling and a hybrid support vector regression (v-SVR) analysis based on supervised machine learning algorithm are used to examine deformation characteristics of surrounding rocks in complex conditions of OP to UG mining transition. These analyses can provide approximation of rock strength properties to be incorporated in the transitions planning optimization problem.

*2.4. Existing Evaluation Tools for Mining Options and Transitions Planning Crown*

Most of the existing optimization models and algorithms for evaluating OP-UG mining options and transitions planning have been incorporated into software packages for easy implementation. These are mostly based on the fundamental evaluation techniques discussed in Section 2.3. Modeling techniques for optimization problems have been discussed in Appendix A. Some of the models and algorithms are Lerchs-Grossman algorithm, Seymour algorithm, floating cone technique, dynamic programming, neural network, theory of graphs, and network flow algorithm. These models and algorithms are used in software packages including NPV Scheduler, Whittle Four-X, MineSched, Vulcan, MineScape, and MineSight [25,26].

GEOVIA Whittle® software which is based on the Lerchs–Grossman algorithm [69] is commonly used to optimize the OP limit prior to assessing the remaining mineral resource outside the pit boundary for underground extraction [7]. Similarly, Blasor pit optimization software which is developed based on a combinatorial mathematical programming model is also used to identify the independent OP mining outline before UG extraction evaluation for remaining mineral resources [9]. Mathematical programming frameworks based on IP, mixed integer linear programming (MILP), and dynamic programming models have been formulated and implemented with commercial optimization solvers for the transitions planning problem [4,17,20,30].

After obtaining the OP outline and thereafter knowing the extent of the UG mining limits, production schedules are generated for each mining option. Software packages based on heuristics and mathematical programming models including GEOVIA Whittle, OptiMine®, and COMET® have been used to produce strategic production schedules for the OP portion of the mine [4,7,17]. Similarly, software packages including Snowden's Evaluator, XPAC®, and OptiMine® have been used to produce the strategic production schedule for the UG portion of the mine [4,7,9,17,18,30].

*2.5. Limitations of Current Models and Algorithms for OP-UG Mining Options and Transitions Planning*

Primary challenges to the mining options optimization problem include the optimization approach used, the exhaustive consideration of contributing variables to the models and algorithms, and geotechnical considerations in defining the transition interface and its contribution to the mining operation. Progressively integrating geotechnical models in strategic mine plans at the prefeasibility stage similar to how geologic models are incorporated will improve the reliability of the mine plan [70,71]. However, the information required to produce a detailed geotechnical model at the prefeasibility stage is limited and therefore difficult to model. Some of the limitations with current models and algorithms for OP-UG mining options optimization include one or more of the following:

1. Consideration of rock support and reinforcement in the optimization process;
2. Consideration of essential infrastructural development in the optimization process;

3. Consideration of stochastic variables;
4. Comprehensiveness and efficiency of models;
5. Optimality assessment.

### 2.5.1. Consideration of Rock Support and Reinforcement in the Optimization Process

Rock support and reinforcement are essential to the stability of openings during the development of an underground mine. The term support generally refers to the various types of geotechnical rock support used to protect underground workers and may include steel mesh, shotcrete, fibrecrete, and a variety of types of steel straps. Reinforcement on other hand refers to the various types of rock reinforcement to help prevent rock movement and may include a variety of types of rockbolts, cablebolts, rebar, and dowel. Cablebolting in stope development can particularly be very costly and may introduce considerable time delays. Most cablebolts need 30 days for the Portland cement grout to properly set for the cablebolts to be fully functional.

Several authors have acknowledged the importance of incorporating geotechnical constraints to the OP-UG transition problem [9,14,15,20,25,47,72]. To verify the impact of geotechnical constraints on the optimal solution, Roberts, Elkington, van Olden, and Maulen [9] recommended that, such constraints need to be incorporated in subsequent studies. Moving beyond determining transition depth and incorporating geotechnical parameters in locating and sizing the transition zone will have direct impact on the feasibility and sustainability of the OP-UG mining project.

These existing models do not integrate the geomechanical classification of the rock formation in the surface-underground mining option and transition problem. Although the geotechnical characteristics of the rock formation are significant to underground mining operations, their added complexities make it difficult to be included in the surface-underground mining option and transition optimization models. According to Bullock [37], mine planning is an iterative process that requires looking at many options and determining which, in the long run, provide the optimum results. Using such iterative process could lead to some inferior solution(s) or sub optimal solution(s) that do not constitute the global optimal solution.

Rock support and reinforcement in the development openings and stopes will increase the operational costs and time (delay the mine life), and further affect the quantity and sequence of rock material extracted from the stopes to the processing plants. To incorporate the rock formation's strength into the formulation, rock mass classification systems [73,74] such as the Geological Strength Index (GSI), Rock Structure Rating (RSR), Rock Mass Rating (RMR), and Q system are determined to characterize the rock formation, and then Kriging applied to populate the block model. According to Abbas and Konietzky [73], these classification systems could be grouped as qualitative or descriptive (e.g., GSI) and quantitative (e.g., Q system, RMR, and RSR) with RMR being more applicable to tunnels and mines. According to Kaiser and Cai [74], data obtained from the geology and geomechanics of the rock formation are fundamental to mine planning and development designs. This is because the behavior of the rock formation varies in the mine and therefore rock mass domaining based on stress data, and geology and geometric data becomes essential. Knowledge of the strength of the rock mass and its behavior are important for the engineering design of all kinds of support for underground excavations [75,76].

### 2.5.2. Consideration of Essential Infrastructural Development in the Optimization Process

The typical UG mine is interspersed with important infrastructure development that ensures the facilitations of the UG mining operations. These infrastructures include but not limited to primary access development, secondary access development, ventilation development, ore pass development, sumps, maintenance and refuge chambers, and fuel station bays. Primary access developments are usually the main development that links the entire UG mine to the surface and could be vertical or inclined shaft(s), decline(s) or adit(s), or tunnel(s). The shaft(s) are often equipped with facilities to transport humans

(workers) and materials for the UG operations. Secondary access development includes the construction of lateral openings such as levels, ore and waste drives, or crosscuts, to link the UG operational activities to the primary access(es) while ventilation development often incorporates a series of bored raises and lateral drive development, and the construction of ventilation controls. Ventilation controls are a range of objects such as regulators, doors, and walls which need considerable time and money to create. Ore pass development entails the construction of vertical or near vertical openings to link the various UG levels to the main ore hoisting station. Similarly, sumps, maintenance and refuge chambers, and fuel station bays are constructed to enable the removal of water being used for the UG operations and the provision of several essential services to operating equipment UG rather than being transported to the surface for such services.

### 2.5.3. Consideration of Stochastic Variables

In recent studies, the industrial practice has been that the production for OP mine and UG mine are independently scheduled and merged into one for the OPUG mine. According to King, Goycoolea, and Newman [18], this approach creates a myopic solution. However, the discussions of the approach were limited to open stoping and extraction sequence. No stochastic variables such as grade and price uncertainty, were included in their model. King, Goycoolea, and Newman [18] further acknowledged that, their methodology in handling the transition problem require further work to handle the applicability, accuracy, and reduction of the optimality gap. Grade uncertainty has been identified to have a significant impact on the determination of the transition point [16].

Although MacNeil and Dimitrakopoulos [20] incorporated grade uncertainty in their work, they further identified some important notable geological uncertainties such as the rock formation, metal content, and relevant rock properties and their impact on the strategic long term planning of a mining project. MacNeil and Dimitrakopoulos [20] further recommended that, financial uncertainty should be incorporated into future studies to improve on their solution method. Ben-Awuah, Otto, Tarrant, and Yashar [4] did not consider uncertainty in their model formulation and further recommended that pre-production capital expenditure (CAPEX) and geological uncertainties should be added to the mining options evaluation. By considering stochastic variables in a risk-based mining options optimization framework, mine plans that can stand the test of time can be generated.

### 2.5.4. Comprehensiveness and Efficiency of Models

Comprehensiveness and efficiency of the models relates to the ability of the optimization framework to exhaustively formulate various scenarios of the OP-UG mining options problem and deliver practical results in a reasonable time frame. Such models are exhaustive and are applicable in a wide range of mining systems. Stacey and Terbrugge [77] indicated that a complete model for handling the transition problem remained a challenge they further reckoned that the transition study for a mine should start at the onset of the mine life and not be deferred to latter years since the planning and implementation could sometimes take up to 20 years for completion. Shinobe [10] developed a software based on a mathematical programming model for this challenge but assumed that the existence of underground reserves have been confined and that their extraction is technically feasible. He later recommended that the results of the program should be viewed only as a preliminary level indication of the economics of underground conversion. No final decision to proceed with the conversion should be taken, solely based on the program's output.

Majority of the current work on the transition problem lack some constraints and solution to a more general problem. This includes consideration of the design capacities and depth as primary indicators for transition in the joint evaluation problem [25]. The approach however was limited in scope in relation to a more generic framework. NPV curves and transition depth for OP to UG mining for the Botuobinskaya pipe deposit were generated to solve the transition problem. From their results, the total NPV of the OPUG mining operation was higher than the standalone NPVs of OP and UG mines for the same

mining depth. In subsequent research, re-handling cost proved to be insignificant when incorporated into the transition model [18]. Additionally, it was observed that there exist undesirable fluctuations in the OPUG production schedules that must be smoothened to achieve a more practical solution. In their work, MacNeil and Dimitrakopoulos [20] incorporated the constraints for mining, processing, metal content, and precedence relationships in their model. According to the constraints affecting the transition problem identified in the works of Opoku and Musingwini [7], those constraints considered are not exhaustive since UG mining capital expenditures were not considered in the model application.

2.5.5. Optimality Assessment

Optimality assessment is a real challenge to current heuristic and meta-heuristic models and algorithms for OP-UG mining options optimization. The underlying optimization approach fundamentally affects the optimality of the resource evaluation. Unless additional steps are put in place to define an upper bound, the extent of optimality of the solution from these models is unknown. Some of the current models can solve the transition problem, usually producing near optimal solutions [14,47]. These existing models and algorithms assume open pit mining operations must surely be an option in the evaluation process and usually precedes the underground mining. Similarly, few models assess the resource with the assumption that underground mining could precede open pit mining. However, a resource assessment that is devoid of the traditional arrangement of the mining options but allows the optimization process to decide the choice and sequence of the mining option(s) is key to achieving global optimal solution. Bakhtavar, Shahriar, and Oraee [2] noted that, few methods (algorithms) have some disadvantages and deficiencies in finding the optimal transition depth. According to Askari-Nasab et al. [78], heuristic algorithms may not produce optimal solutions. This could lead to loss of huge financial benefits resulting from implementing sub-optimal mine plans.

Finch [47] also highlighted that, the effort of producing OPUG mining schedules for the possible candidates of transition interface for all the various mining and processing capacities could be time consuming and costly. The process commonly leads to the generation of sub-optimal solutions since the problem is not thoroughly investigated. According to Richard and Stefan [79], designs optimized for deterministic cases are often sub-optimal when uncertainties are recognized and their effects understood. By applying mathematical programming models with current high computing resources [31], optimal or near optimal solutions with known optimality gap can be obtained for the OP-UG mining options optimization problem in a practical time frame and at an acceptable computational cost.

## 3. Summary and Conclusions

The problem of optimizing resource exploitation depends largely on the mining option used in the extraction. Some mineral deposits extend from the near surface to several meters in depth. Such deposits can be amenable to both surface mining and/or underground mining, and this leads to the surface–underground mining options and transitions optimization. Relevant literature review on the SUMOTO problem has been conducted and documented. Research works in this area have primarily focused on different variations of determining the transition depth between open pit and underground mines, and the subsequent optimization of the strategic schedule for each mining option. Heuristics and exact solution methods have both been used in the past to attempt the SUMOTO problem. An algorithm or model that comprehensively and simultaneously determines an optimized open pit mine, transition interface and an underground mine for any orebody by both surface and underground mining methods in a single run will significantly add value to the mining industry. A matrix showing the various approaches adopted by researchers in tackling the OP-UG transition problem in the last decade has been developed (Table 1).

Shortfalls on previous research for mining options optimization have been discussed and opportunities for further studies identified. Notable limitations of current models

and algorithms for the SUMOTO problem include: (1) consideration of rock support and reinforcement in the optimization process; (2) consideration of essential infrastructural development in the optimization process; (3) consideration of stochastic variables; (4) comprehensiveness and efficiency of models; and (5) optimality assessment of model solution. These identified limitations may often lead to sub-optimal global solution to the SUMOTO problem thereby affecting the viability of the mining project. It is therefore essential to develop a rigorous optimization framework that attempts to address some of these deficiencies.

Although the main sources of uncertainties in mining options studies have been found to include financial, technical, and geological, research on strategic mining options have handled these uncertainties independently. Incorporation of geological uncertainties in current strategic mining options studies have been applied in different forms, including, grade and tonnage uncertainties, probability indices, and the use of algorithms to further define these uncertainties. The incorporation of financial uncertainties together with geological uncertainties is limited in current research on mining options studies. As uncertainties cannot be eliminated in the mining options problem, the best strategy is to quantify uncertainty, reduce uncertainty and manage the associated risk during the production scheduling process.

In the last decade, different variations of mathematical programming models have been used by researchers to solve challenges associated with the surface-underground mining options problem. The main variations are either deterministic (linear programming and integer and mixed-integer programming) or stochastic (dynamic programming and stochastic programming) or combination of both.

## 4. Recommendations

The authors conclude by proposing further research into the formulation of an integrated stochastic programming model for the mining options and transitions optimization problem for all deposits including base and critical minerals. Figure 4 is a representation of the recommended research considerations to solve the complexities associated with the strategic surface-underground mining options and transitions optimization problem.

To add significant value to the mining industry, the proposed research approach in Figure 4 will result in a stochastic mining options and transitions optimization model that has the capacity to generate strategic mining options including: OP mining, UG mining, simultaneous OPUG mining, OP mining followed by UG mining, and UG mining followed by OP mining. The proposed model should be applicable at the prefeasibility stage of mining project to guide mine planners and investors in making important decisions. Some of the performance indicators of the model should include net present value, internal rate of return, discounted cashflow, price to cost ratio, blending ratio, production smoothness, stripping ratio, mine life, mining recovery, and average run-of-mine (ROM) grade. The proposed model should have the features and capacity to:

(a)  Take in simulated block models as inputs in a risk-based or stochastic framework that considers grade, price, and cost uncertainty;

(b)  Evaluate strategic mining options with exhaustive constraints for large-scale mining projects through efficient numerical modeling and computational techniques. In addition to standard mining and technical constraints, other notable constraints include controls for safety, geotechnical, geological, and hydrogeological conditions of the mining area;

(c)  Integrate waste management constraints and synergies wherein waste material and tailings from open pit mining can be used for underground backfilling; as well as characterizing mineralized waste material as future resource;

(d)  Leverage resource governance and synergies whereby open pit low grade ore can be blended with underground high-grade ore to improve processing recovery and extend mine life;

(e)  Determine the size and capacity of the mining project using real value options approach.

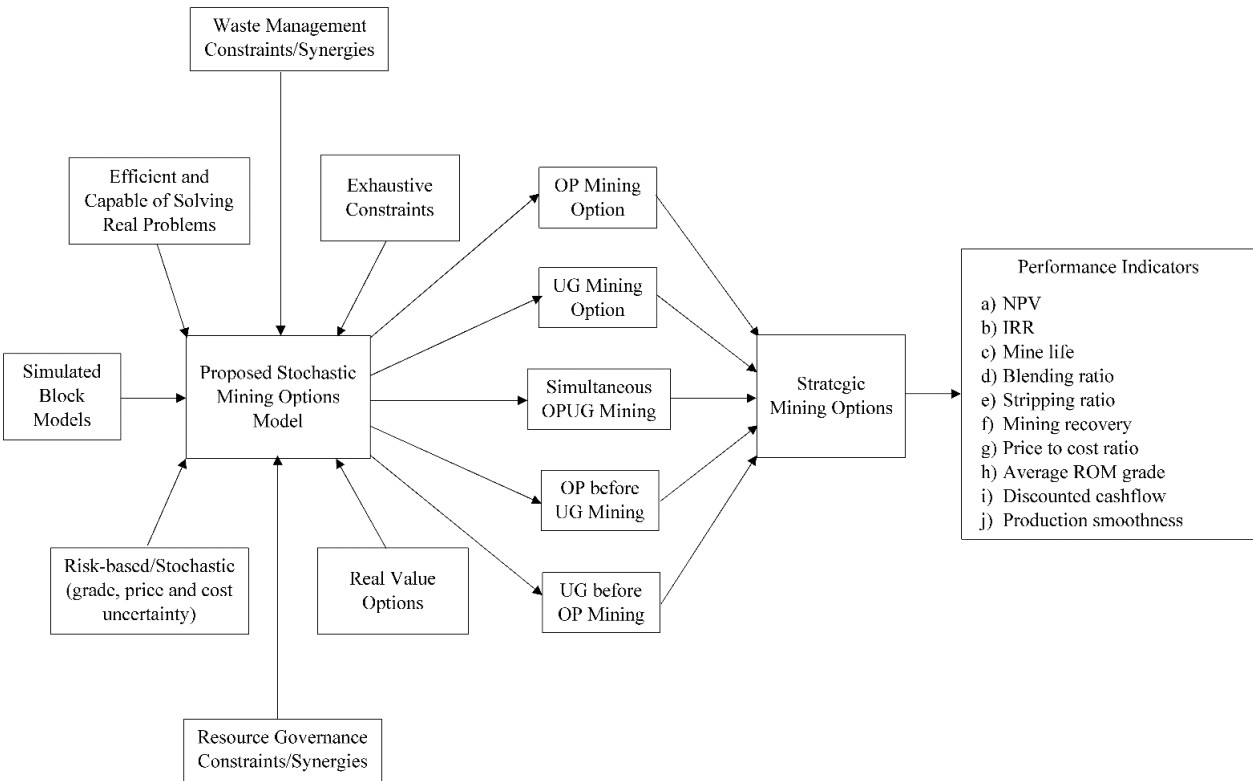

**Figure 4.** Schematic representation of the proposed research approach for strategic OP-UG mining options optimization.

**Author Contributions:** The research work was completed by B.O.A., under the supervision of E.B.-A. All authors have read and agreed to the published version of the manuscript.

**Funding:** This work was supported by the Ontario Trillium Scholarship Program, IAMGOLD Corporation and Natural Sciences and Engineering Research Council of Canada [DG #: RGPIN-2016-05707; CRD #: CRDPJ 500546-16].

**Institutional Review Board Statement:** Not applicable.

**Informed Consent Statement:** Not applicable.

**Data Availability Statement:** Data sharing not applicable. No new data were created or analyzed in this study. Data sharing is not applicable to this article.

**Conflicts of Interest:** The authors declare no conflict of interest.

## Appendix A. Modeling Techniques for Optimization Problems

The literature review conducted on the surface-underground mining options and transitions optimization (SUMOTO) problems indicate that mining options and transitions optimization planning are modeled and solved using different evaluation techniques. Understanding the various classifications of optimization models and their advantages are essential to the development of an optimization framework capable of handling the notable limitations and gaps associated with the current models and algorithms for SUMOTO problems. These optimization models may be grouped into four broad types according to how well they are able to define a given problem: (1) operational exercise models, (2) gaming models, (3) simulation models, and (4) analytical or mathematical programming models. Simulation and analytical models are widely used for mining optimization problems due to their practicality, robustness, and efficiency [80,81]. Table A1 shows classification of certainty and uncertainty models.

**Table A1.** Classification of certainty and uncertainty models [23].

| Class | Strategy Evaluation | Strategy Generation |
|---|---|---|
| Certainty | Deterministic simulation<br>Econometric models<br>Systems of simultaneous equations<br>Input-output models | Linear programming<br>Network models<br>Integer and mixed integer programming<br>Nonlinear programming<br>Control theory |
| Uncertainty | Monte Carlo simulation<br>Econometric models<br>Stochastic processes<br>Queueing theory<br>Reliability theory | Decision theory<br>Dynamic programming<br>Inventory theory<br>Stochastic programming<br>Stochastic control theory |

*Appendix A.1. Operational Exercise Models*

Operational exercise is a modeling method that relates variables to the actual environment where the study decision will be applied [23]. This method has the highest degree of realism when compared to other techniques of modeling. It is usually prohibitive, expensive to implement, and associated with extreme challenge when alternatives must be assessed. This then leads to sub-optimization of the final solution. A human decision-maker forms part of the modeling processes of operational exercises.

*Appendix A.2. Gaming Models*

Gaming is a modeling approach constructed to represent more simple or abstract entities in the real world [23]. This method gives the mine planner the opportunity to try several variations of the decisions to make. A human decision-maker is part of the modeling processes for gaming models.

*Appendix A.3. Simulation Models*

Simulation models provide several ways to assess the performances of alternatives outlined by the planner. They do not allow much interferences from human interactions during the computational analyses stage of implementation [23]. Simulation models could sometimes be compared to gaming models. However, they involve the application of logical arithmetic performed in a particular sequence usually by computer programs. If the problem is exclusively defined in an analytical form, much flexibility and realistic results are attained. This is essential if uncertainties are critical components of the decisions being made. A human decision-maker is external or not part of the modeling processes of simulation and analytical models.

*Appendix A.4. Analytical or Mathematical Programming Models*

The fourth model category is analytical models. These models represent the problem completely in mathematical forms, usually by means of a criterion or objective subject to series of constraints that impact on the decision being made [23]. The mathematical formulation helps to determine the optimal solution in the presence of the set of constraints. Although mathematical models highly simplify the problem, they are less expensive and easy to develop.

Mathematical programming is a significant method when decision variables must be quantified for planning. The problem is defined with mathematical expressions in a well-defined structure to find an optimal solution based on performance evaluation criteria including cost, profit, and time. The optimal solution is obtained from a feasible region of alternative results. The expressions are parameters (input data) and variables which represent the optimization results or outcome. When multiple criteria decisions are required for any complex problem, a multi-objective mathematical programming

model involving more than one objective function is deployed simultaneously. The main advantages of mathematical programming models (MPMs) are: (a) comparatively simple with high approximations of complicated problems, and (b) ability to search the feasible solution spaces among competing variables and alternatives [82]. Common MP techniques are stochastic programming, deterministic programming, dynamic programming, LP, nonlinear programming, IP, and MILP.

### Appendix A.4.1. Stochastic Programming

Stochastic programming is a special case of programming in which some of the constraints or parameters depend on random variables. This type of programming is used to solve problems that involve uncertainty and allow stochastic variables to be accounted for [82].

### Appendix A.4.2. Deterministic Programming

Deterministic programming is a form of MP that is rigorous and solves problems in finite time. The optimization model is deterministic when the parameters considered in the model are known constants [23]. The method is useful when global solution is required and is extremely difficult to find a feasible solution.

### Appendix A.4.3. Dynamic Programming

Dynamic programming is a form of mathematical programming used to solve multistage complicated problems [82]. A large-scale problem is simplified by breaking it down into simpler sub-problems in a nested recursive manner as opposed to deterministic programming.

### Appendix A.4.4. Linear Programming

This is a distinct case of convex programming where the model objective and the equality and inequality constraints are expressed as linear functions. The feasible solution set is usually a polytope or polyhedron with connected sets of polygonal faces and convex. The optimal solution could be a single vertex, edge or face, or even sometimes the entire feasible region when dealing with high dimension problems. Typical LP solutions could arrive as infeasible or unbounded [83].

### Appendix A.4.5. Nonlinear Programming

Nonlinear programming is a special case of programming which could be convex or non-convex with nonlinear objective functions or nonlinear constraints or [83]. Some assumptions are often made on the shape and function behavior when solving nonlinear programming problems [23]. Nonparametric and simulation-based optimization techniques have been used to explicitly solve models with nonlinear constraints. However, computational feasibility have been a major disadvantage in the development and implementation of MPMs [82].

### Appendix A.4.6. Integer Programming

Integer programming (IP) is a special type of LP where some or of all the decision variables are constrained to take on integers and therefore not continuous. If all the variables are discrete or integers or binaries, the model is referred to as pure integer programming. Essentially, IP problems are hard problems because they are difficult to solve and therefore referred to as combinatorial analysis than LP [23].

### Appendix A.4.7. Mixed Integer Linear Programming

This is a special form of IP in which some decision variables are constrained to take on integers and others continuous. Continuous variables indicate that the variables could take on fractions as part of the solution regime. Due to the strength of MILP formulations, it is proposed as the model for solving the SUMOTO problem in this research. The MILP

formulation structure is well defined with objective function, and a set of constraints to achieve the mining decisions that are usually made up of continuous variables and integers.

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
