# Peer review of "A Review of Models and Algorithms for Surface-Underground Mining Options and Transitions Optimization: Some Lessons Learnt and the Way Forward"

_mining, doi:10.3390/mining1010008_

Round 1

Reviewer 1 Report

Dear authors, thanks for your article A Review of Models and Algorithms for Surface-Underground  Mining Options and Transitions Optimization: Some Lessons  Learnt and the Way Forward. The title of the article corresponds to its content. This is a review article. But most of the text is a description of already known methods. There are no references to the works of scientists working in countries where the mining industry is developed. 

Author Response

A list of mining companies with references have already been provided on L233, L234 and L235. The statement, “Lac Des Iles mine located in Canada and Newmont Ahafo mine in Ghana both operate OPUG mining operations” is added on L235 and L236 to enhance the paper.

Reviewer 2 Report

This paper reviews the methods of planning for open pit - underground transition. I think the paper is quite well-written and gives a nice summary of the topic. As a review paper, I would see it as of additional value if the authors could make some more focused insights in the conclusions section BASED on the review they have done (see comments below). This would improve the overall value of this contribution (gave two out of five stars). 

MAJOR COMMENTS: 
- Chapter 3 (L659-758): I think this is somewhat detached from the other text. Maybe you should consider presenting this in the appendix
- L811-832: these are very general considerations. I would suggest ending your paper with some more insights that are of value to the reader

MINOR COMMENTS: 
- L283-284: "It is based on the premise that, UG mines are characterized by high cut-off grades and therefore have higher cash flow compared to OP mines for the same throughput." 
* I think should be "per an equivalent unit of throughput", because absolute cash flows do not take into account the scale of mining and therefore it does not mean that UG mines produce higher cashflows when measured in ABSOLUTE units
- L351, 354, 481: Reference-errors in pdf
- Table 1: explain abbreviations, e.g. "MILP", "COMET", "MRO",...
* It seems a bit surprising that the newer algorithms are more constrained in terms of section "Outputs/Performance". Please explain and/or consider if the year of publication is the best way to arrange these
- L452, 453: explain abbreviations IP, LP

Author Response

Responses to Reviewer 2:

This paper reviews the methods of planning for open pit - underground transition. I think the paper is quite well-written and gives a nice summary of the topic. As a review paper, I would see it as of additional value if the authors could make some more focused insights in the conclusions section BASED on the review they have done (see comments below). This would improve the overall value of this contribution (gave two out of five stars).

Chapter 3 (L659-758): I think this is somewhat detached from the other text. Maybe you should consider presenting this in the appendix.

Response: Chapter 3 has been moved to appendix

L811-832: these are very general considerations. I would suggest ending your paper with some more insights that are of value to the reader.

Response: L811-833 has been replaced with discussions on Figure 4 highlighting the proposed research approach that will bring a step-change to new models. Figure 4 has been replaced with an improved one.

L283-284: "It is based on the premise that, UG mines are characterized by high cut-off grades and therefore have higher cash flow compared to OP mines for the same throughput." I think should be "per an equivalent unit of throughput", because absolute cash flows do not take into account the scale of mining and therefore it does not mean that UG mines produce higher cashflows when measured in ABSOLUTE units.

Response: The following phrase “per an equivalent unit of throughput,” has been added to the sentence to enhance the understanding of the whole statement on L283.

L351, 354, 481: Reference-errors in pdf.

Response: “Table 1” is inserted on the error messages on L351 and L354. “2.3” is inserted on the error message on L481.

Table 1: explain abbreviations, e.g. "MILP", "COMET", "MRO",... It seems a bit surprising that the newer algorithms are more constrained in terms of section "Outputs/Performance". Please explain and/or consider if the year of publication is the best way to arrange these.

Response: MILP is explained on L492. COMET is not an abbreviated word. MRO is already explained on L58. NPV is explained on L177, IRR is explained on L178. Explanation to the selection of “Outputs/Performance” is already done on L361 to L362.

Using the year of publication to arrange the past research was to provide a focus on progression of models and algorithms developed for OP-UG mining options over the years as well as their limitations.

L452, 453: explain abbreviations IP, LP.

Response: Integer programming (IP) is added on L452. LP is already explained on L450

Reviewer 3 Report

The paper deals with an overview of mining methods and procedures and their optimization in surface and underground mines. The authors tried to summarize an overview of the methods used and then evaluate them.
The mining of minerals (especially underground) is usually a financially demanding investment, and for this reason, it is necessary to optimize the mining process in terms of maximizing yields.
In the case of underground mining, it is necessary to take into account a number of other factors such as geological and hydrogeological conditions, mining and technical conditions, and geotechnical conditions.
It would also be appropriate to standardize the methods in relation to the mined raw material, taking into account specific conditions, e.g. in deep coal mines (danger of explosion of coal dust and methane).
The authors sought a comprehensive solution to the issue of mining, focusing their attention on high economic profitability (Table 1 Commodity, especially Gold). At present, however, the need for mining is focused primarily on the so-called critical raw materials needed for the electrical engineering industry and the fast-growing battery industry.
It might be appropriate to supplement the breakdown of methods with an assessment of the life cycle of the mine.

Author Response

Responses to Reviewer 3:

Comments: The paper deals with an overview of mining methods and procedures and their optimization in surface and underground mines. The authors tried to summarize an overview of the methods used and then evaluate them. The mining of minerals (especially underground) is usually a financially demanding investment, and for this reason, it is necessary to optimize the mining process in terms of maximizing yields. In the case of underground mining, it is necessary to take into account a number of other factors such as geological and hydrogeological conditions, mining and technical conditions, and geotechnical conditions. It would also be appropriate to standardize the methods in relation to the mined raw material, taking into account specific conditions, e.g. in deep coal mines (danger of explosion of coal dust and methane).

Response: The following statement has been added to L843-845 to highlight the additional constraints required for the proposed model:

“In addition to standard mining and technical constraints, other notable constraints include controls for safety, geotechnical, geological and hydrogeological conditions of the mining area”

Comments: The authors sought a comprehensive solution to the issue of mining, focusing their attention on high economic profitability (Table 1 Commodity, especially Gold). At present, however, the need for mining is focused primarily on the so-called critical raw materials needed for the electrical engineering industry and the fast-growing battery industry.

Response: Paragraph 1 of Section 4: Recommendations has been updated as follows:

“The authors conclude by proposing further research into the formulation of an integrated stochastic programming model for the mining options and transitions optimization problem for all deposits including base and critical minerals”.

Comments: It might be appropriate to supplement the breakdown of methods with an assessment of the life cycle of the mine.

Response: Performance indicators like the life of mine and mining recovery are included in Table 1 and the recommended research consideration in Figure 4 to assess the life cycle of the mine and resource depletion ratio of the various applications.

Reviewer 4 Report

The article  in present form is ready to publish.

Author Response

Responses to Reviewer 4:

comment: The article in present form is ready to publish.

Response: Thank you
